# Re-Thinking Inverse Graphics With Large Language Models

**Peter Kulits***                                                    *kulits@tue.mpg.de*
*Max Planck Institute for Intelligent Systems, Tübingen, Germany*

**Haiwen Feng***                                                    *hfeng@tue.mpg.de*
*Max Planck Institute for Intelligent Systems, Tübingen, Germany*

**Weiyang Liu**                                                    *wl396@cam.ac.uk*
*Max Planck Institute for Intelligent Systems, Tübingen, Germany, University of Cambridge*

**Victoria Abrevaya**                                                    *vabrevaya@tue.mpg.de*
*Max Planck Institute for Intelligent Systems, Tübingen, Germany*

**Michael J. Black**                                                    *black@tue.mpg.de*
*Max Planck Institute for Intelligent Systems, Tübingen, Germany*

**Reviewed on OpenReview:** *https://openreview.net/forum?id=u0eiu1MTS7*

## Abstract

Inverse graphics – the task of *inverting* an image into physical variables that, when rendered, enable reproduction of the observed scene – is a fundamental challenge in computer vision and graphics. Successfully disentangling an image into its constituent elements, such as the shape, color, and material properties of the objects of the 3D scene that produced it, requires a comprehensive understanding of the environment. This complexity limits the ability of existing carefully engineered approaches to generalize across domains. Inspired by the zero-shot ability of large language models (LLMs) to generalize to novel contexts, we investigate the possibility of leveraging the broad world knowledge encoded in such models to solve inverse-graphics problems. To this end, we propose the Inverse-Graphics Large Language Model (*IG-LLM*), an inverse-graphics framework centered around an LLM, that autoregressively decodes a visual embedding into a structured, compositional 3D-scene representation. We incorporate a frozen pre-trained visual encoder and a continuous numeric head to enable end-to-end training. Through our investigation, we demonstrate the potential of LLMs to facilitate inverse graphics through next-token prediction, without the application of image-space supervision. Our analysis enables new possibilities for precise spatial reasoning about images that exploit the visual knowledge of LLMs. We release our code and data at `https://ig-llm.is.tue.mpg.de/` to ensure the reproducibility of our investigation and to facilitate future research.

## 1 Introduction

The formulation of vision as "inverse graphics" traces its roots back at least to Baumgart (1974) (see also Knill et al. (1996) and Yuille & Kersten (2006)). While the term encompasses various ideas and approaches to vision problems, it is often equated with "analysis by synthesis" (Grenander, 1976–1981). What is typically meant here, however, is more akin to model fitting. This generally presupposes that one has models of the world, knows roughly where they are in the scene, and then fits them to image evidence.

A more strict interpretation of inverse graphics targets the creation of a *graphics program* (Ritchie et al., 2023): a structured representation that can be used by a rendering engine to approximately reproduce a

---

*Co-first author

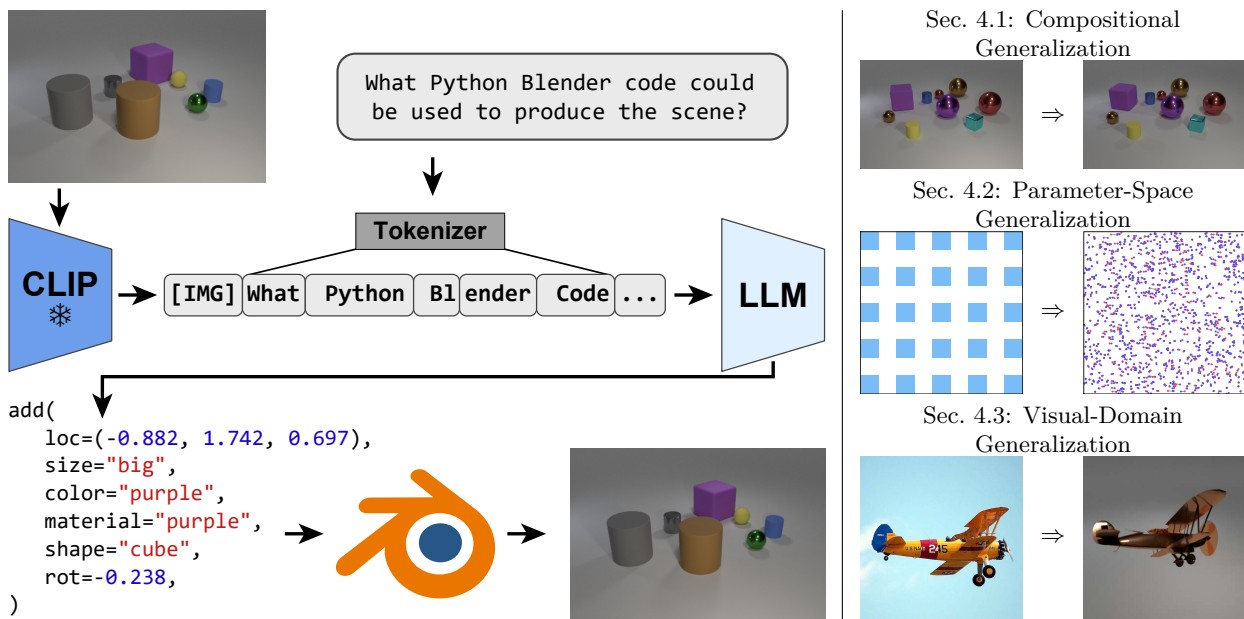

Figure 1: **IG-LLM.** We present the Inverse-Graphics Large Language Model (IG-LLM) framework, a general approach to solving inverse-graphics problems. We instruction-tune an LLM to decode a visual (CLIP) embedding into graphics code that can be used to reproduce the observed scene using a standard graphics engine. Leveraging the broad reasoning abilities of LLMs, we demonstrate that our framework exhibits natural generalization across a variety of distribution shifts without the use of special inductive biases.

3D scene. These programs are compact and interpretable abstractions of visual primitives (Wu et al., 2017; Jones et al., 2023), thereby aiding scene comprehension (Wu et al., 2017; Yi et al., 2018). The objective extends beyond mere pixel- or object-level interpretation of an image; it seeks to leverage the inherent spatial and physical relationships among objects that are essential for holistic scene understanding.

Our goal is to generate such a graphics program from a single image capable of reproducing the 3D scene and its constituent objects using a traditional graphics engine. This approach, known as visual program induction, has garnered significant attention, encompassing works on a variety of problem domains. Wu et al. (2017) propose the concept of "neural scene de-rendering," wherein custom markup code, translatable into renderer-friendly inputs, is inferred from images. While their method can handle synthetic scenes featuring arbitrarily many objects, it grapples with generalization beyond its training-data distribution. Subsequent research (Yi et al., 2018) explores the utility of graphics programs for the downstream task of visual question answering (VQA). However, their scene-reconstruction method still struggles with generalization, particularly regarding objects with unseen attribute combinations (e.g., a known shape with a novel color–shape combination).

To address the problem of generalization, a method must possess a deep understanding of the visual world and its physical properties. Here, we explore whether we can exploit the generalization abilities of large language models (LLMs) for this purpose. LLMs have demonstrated remarkable performance across a wide variety of vision–language tasks, ranging from producing detailed textual descriptions of images (Alayrac et al., 2022) and generating realistic images from text (Koh et al., 2023), to tasks such as visual question answering (Ouyang et al., 2022; OpenAI, 2023), visual instruction following (Liu et al., 2023; Zhu et al., 2023), and robot planning (Huang et al., 2022; Singh et al., 2023). Intriguingly, these models are designed with generic architectures and are initially trained with objectives that are not specifically tailored to a downstream task. The breadth of their training data endows them with the capacity for compositional reasoning about the world. However, their proficiency in conducting *precise spatial reasoning* within the 3D Euclidean world remains largely unexplored. This prompts the question: Can LLMs, originally used to address *semantic*-level queries, be applied to the *precise* realm of inverse-graphics tasks? And if so, how?

To address this question, we investigate the potential of LLMs to perform such tasks. We hypothesize that LLMs can be trained with simple demonstrations to perform precise inverse-graphics reasoning. This idea

draws inspiration from instruction tuning (Taori et al., 2023; Chung et al., 2024) in the language-processing domain, where LLMs acquire instruction-following skills after being fine-tuned on a limited set of curated training samples. We anticipate that LLMs, endowed with broad knowledge about the physical world, can be taught to recover accurate graphics programs from images beyond their training distribution. This insight motivates a reevaluation of the conventional inverse-graphics pipeline, leading to the proposal of a new LLM-based framework: the Inverse-Graphics Large Language Model (*IG-LLM*). We fine-tune an LLM equipped with a pretrained text-aligned visual encoder, using an instruction-based synthetic dataset, and explore the model's capacity to infer graphics programs with accurate estimates of object quantity, shape, size, color, material, location, and orientation, as illustrated in Fig. 1.

However, a question arises regarding the suitability of LLMs – and natural language – for generating the precise measurements necessary for inverse graphics, given the discrete nature of their token-based output. This constraint poses challenges for reasoning within metric spaces such as Euclidean space. To address this, we explore the integration of a *numeric head* in the language-based output (see Fig. 2b), where numbers are represented as continuous values rather than discrete-token sequences. We compare this approach and observe across evaluation that it achieves improved precision and an expanded generalization capacity.

Our study is an examination of the adaptability of LLMs to novel domains and an attempt to understand how these powerful, semantically driven models can be repurposed and refined to gain a precise metric understanding of the 3D world. While our investigation is preliminary, our work paves the way for further endeavors to capitalize on the rapid advancements in LLMs.

## 2 Related Work

**Visual Program Induction.** Visual program induction is a subfield of program synthesis that is focused on recovering a *graphics* program from a given visual target (Gulwani et al., 2017). Graphics programs, also known as procedural or symbolic programs, offer a concise, structured, and interpretable representation for scenes and have garnered significant attention in the field – see Ritchie et al. (2023) for an in-depth overview. Commonly employed program types include constructive-solid geometry (CSG) (Du et al., 2018; Kania et al., 2020; Ren et al., 2022; Sharma et al., 2018a; Yu et al., 2022), computer-aided design (CAD) Ganin et al. (2018); Li et al. (2020a; 2022a); Seff et al. (2022); Xu et al. (2021), vector graphics (e.g., SVG) (Reddy et al., 2021a;b), and L-systems (Guo et al., 2020), as well as custom program domains (Ellis et al., 2018; Tian et al., 2019; Deng et al., 2022; Hu et al., 2022b). Program discovery can be achieved from simplified representations of the same modality, such as 2D hand drawings or synthetic patterns (Št'ava et al., 2010; 2014; Sharma et al., 2018a; Ellis et al., 2019; Riso et al., 2022; Ganeshan et al., 2023; Seff et al., 2022), as well as 3D meshes and voxels (Ganeshan et al., 2023; Jones et al., 2022; Tian et al., 2019; Bokeloh et al., 2010; Willis et al., 2021; Sharma et al., 2018a; Ellis et al., 2019). There have also been efforts in recovering 3D scenes from natural 2D images using graphics programs (Mansinghka et al., 2013; Kulkarni et al., 2015; Wu et al., 2017; Yi et al., 2018; Mao et al., 2019; Liu et al., 2019; Li et al., 2020b; Gothoskar et al., 2021; Ganin et al., 2018; Kar et al., 2019; Devaranjan et al., 2020). Kulkarni et al. (2015) propose a probabilistic programming language for representing arbitrary 2D/3D scenes, demonstrating preliminary results for analysis of faces, bodies, and objects. Wu et al. (2017) infer custom-designed markup code from images that can be easily translated to renderer-friendly inputs. The work can handle scenes with a number of objects but cannot generalize beyond the training-data distribution. In follow-up work, Yi et al. (2018) investigate how graphics programs can be used for visual question answering (VQA). However, their scene reconstruction also struggles with generalization problems, particularly with unseen attribute combinations. Liu et al. (2019) present a new language for representing scenes, along with a hierarchical approach for inference. Meta-SIM (Kar et al., 2019; Devaranjan et al., 2020) uses probabilistic scene grammars to recover synthetic scenes from natural images, which are then used to train a generative scene-synthesis model. Despite promising results, such methods require special training data and complex modular architectures, and are difficult to generalize beyond their training distribution.

**Vision as Inverse Graphics.** Dating back to Larry Roberts's Blocks-World thesis (Roberts, 1963), there is a long history of work that treats computer vision as the inverse problem to computer graphics. Efforts have included estimating object pose (Lepetit et al., 2009; Tejani et al., 2014; Pavlakos et al., 2017; Xiang et al.,

2018; Wang et al., 2021b; 2019; 2021a; Ma et al., 2022; Labbé et al., 2020) and reconstructing shape (Choy et al., 2016; Fan et al., 2017; Groueix et al., 2018; Mescheder et al., 2019; Wang et al., 2018; Sitzmann et al., 2019; Park et al., 2019) from single images. Multi-object scenes have also been recovered via geometric approaches (Gkioxari et al., 2019; Denninger & Triebel, 2020; Shin et al., 2019), but they often overlook the semantics and interrelation between objects, hindering further reasoning. Holistic 3D-scene understanding takes the approach a step further by reconstructing individual objects together with the scene layout. Early methods focus on estimating 3D bounding-box representations (Hedau et al., 2009; Lee et al., 2009; Mallya & Lazebnik, 2015; Ren et al., 2017; Dasgupta et al., 2016), whereas more-recent works emphasize the reconstruction of finer shapes (Zhang et al., 2021; Liu et al., 2022; Gkioxari et al., 2022) along with instance segmentations (Kundu et al., 2022; Yao et al., 2018; Dahnert et al., 2021; Nie et al., 2020; Zhang et al., 2023b; Nie et al., 2023). Closely related are methods that perform CAD or mesh-model retrieval followed by 6-DoF pose estimation of individual objects (Aubry et al., 2014; Bansal et al., 2016; Lim et al., 2014; Tulsiani & Malik, 2015) or scenes (Izadinia et al., 2017; Huang et al., 2018; Salas-Moreno et al., 2013; Kundu et al., 2018; Gümeli et al., 2022; Kuo et al., 2020; 2021; Engelmann et al., 2021). An alternative to detailed shape reconstruction is *primitive* reconstruction, where objects or scenes are explained by a limited set of geometric primitives, offering a higher level of abstraction. This direction has been studied extensively (Roberts, 1963; Binford, 1975; Hedau et al., 2009; Gupta et al., 2010a;b), and it is still actively researched (van den Hengel et al., 2015; Tulsiani et al., 2017; Paschalidou et al., 2019; 2020; 2021; Deng et al., 2020; Kluger et al., 2021; Monnier et al., 2023; Vavilala & Forsyth, 2023). While these works typically produce accurate reconstructions, they involve complex pipelines with multiple modules and require special training data, limiting generalization under distribution shifts. In contrast, we explore the use of LLMs as a potentially simpler and more-generalizable solution to the inverse-graphics problem.

**LLMs and 3D Understanding.** Recent and concurrent efforts have explored the use of LLMs for 3D-related tasks, including 3D question answering (Hong et al., 2023; Dwedari et al., 2023), navigation (Hong et al., 2023; Zhang et al., 2023a), text-to-3D (Sun et al., 2023), procedural-model editing (Kodnongbua et al., 2023), and multi-modal representation learning (Xue et al., 2023; Hong et al., 2023). To the best of our knowledge, our work is the first to investigate the application of LLMs to inverse-graphics tasks.

## 3 Method

The goal of this work is to assess the efficacy of LLMs in inverse-graphics tasks. We frame the problem as that of estimating a graphics program from a single image (see Fig. 1) and fine-tune an LLM using a small, synthetic dataset. We begin by analyzing the advantages this approach offers over traditional approaches in Sec. 3.1. Subsequently, we describe the details of our methodology in Sec. 3.3. Finally, we elaborate on the design and motivation of the *numeric head* to enable precise metric reasoning in Sec. 3.4.

### 3.1 Traditional Neural Scene De-Rendering

Our approach builds upon the concept of neural scene de-rendering introduced by Wu et al. (2017). In this framework, the goal is to develop a generalizable model capable of comprehensively understanding a scene by estimating a graphics program executable by a renderer. NS-VQA (Yi et al., 2018) is representative of this paradigm, where the task is addressed by decomposing the visual input using multiple modules with task-specific visual inductive biases. The method includes several components: a region-proposal network for object detection, a segmentation network for isolating the objects from the background, an attribute network for classifying various discrete graphics attributes, and a localization network for predicting the object's spatial location. Each network is independently trained in a supervised manner with a specific objective, and their outputs are aggregated to produce a structured representation of the scene.

### 3.2 What Can LLMs Offer?

The broad success of LLMs can be largely attributed to their exceptional ability to generalize. Unlike models that rely on task-specific inductive biases or well-crafted training objectives, LLMs (Brown et al., 2020; Radford et al., 2019; Touvron et al., 2023) perform proficiently across a variety of language tasks with

relatively minor design differences and a simple training approach. This success can be attributed to the scale of the models and the sets of in-the-wild data on which they are trained.

A particularly interesting development in recent years has been the adaptation of LLMs to downstream tasks through instruction tuning (Chung et al., 2024; Wang et al., 2023; Chiang et al., 2023), where LLMs are fine-tuned on a small set of curated task-specific training samples (e.g., the 52K instruction-following examples in Stanford Alpaca (Taori et al., 2023)). This suggests a paradigm shift from traditional approaches, where generalization is often attained by scaling the amount of task-specific training data. LLMs are primarily trained via an unsupervised next-token-prediction objective to perform language-completion tasks, thereby unifying various natural language processing (NLP) tasks within a generic framework. Our research aims to explore whether this generalized approach can be effectively extended to the task of scene de-rendering while preserving their strong generalization capabilities.

### 3.3 Tuning LLMs for Inverse Graphics

While LLMs are traditionally trained to complete language-token sequences, VQA works (Alayrac et al., 2022; Li et al., 2022b; Liu et al., 2023) have demonstrated that large pretrained vision transformers (Dosovitskiy et al., 2021; Radford et al., 2021) can be efficiently adapted as visual tokenizers. Such works unify image and language understanding, interleaving visual embeddings with language tokens for the LLM. In line with this approach, we adopt a similar strategy, constructing an LLM capable of "seeing" the input image and returning a structured code representation of the input scene.

In the subsequent paragraphs, we detail the base architecture, elucidate on the process of vision–language alignment, and introduce our methodology for preparing synthetic data for visual instruction fine-tuning. A high-level overview of our pipeline can be seen in Fig. 1.

**Architecture.** Our model is based on an instruction-tuned variant (Peng et al., 2023) of LLaMA-1 7B (Touvron et al., 2023) in conjunction with a frozen CLIP (Radford et al., 2021) vision encoder, serving as the visual tokenizer. We apply a learnable linear projection to link the vision embeddings with the word-embedding space of the LLM.

**Vision–Language Alignment.** The linear vision-encoder projection is initially trained using the feature-alignment pre-training method from LLaVA (Liu et al., 2023). This training uses instruction sequences constructed from image–caption pairs sourced from the Conceptual Captions dataset (CC3M) (Sharma et al., 2018b). The LLM receives the projected image embedding, followed by a randomly sampled directive tasking the model to describe the image and its corresponding caption. Throughout this stage, all model parameters remain fixed except for the learnable vision projector. To ensure the generality of our model, we refrain from additional instruction tuning following this initial feature alignment.

**Training-Data Generation.** CLEVR (Johnson et al., 2017) is a procedurally generated dataset of simple 3D objects on a plane. The primitives are assigned randomly sampled attributes such as shape (sphere, cube, and cylinder), size, color, material, and spatial pose. Shape, size, color, and material are discrete attributes, while pose is a continuous parameter specifying the object's location and orientation. The images from the sampled scenes are rendered using the Blender (2018) modeling software from its Python-scripting API. Our domain-specific language consists of `add` functions, facilitating the insertion of objects with specified attributes into the scene using Blender. See Fig. 1 for an example of the representation of a single object.

To train our model, we generate pairs of rendered images and their corresponding code, prompting the model with the rendered image followed by the question, "What Python Blender code could be used to produce the scene?" The model is then trained with a standard next-token prediction objective (Bengio et al., 2000), aiming to maximize the conditional probability of the next token given the previous ones:

$$p(x) = \prod_{i=1}^{n} p\left(s_i | s_1, \ldots s_{i-1}\right), \tag{1}$$

where $s_i$ represents the $i$th token. Numbers are rendered with three decimal places in the text-based (tokenized) training data. We order the `add` statements front-to-back in the objective token sequence and shuffle the order of the object attributes. See Figs. S.2 to S.6 for complete examples of each task.

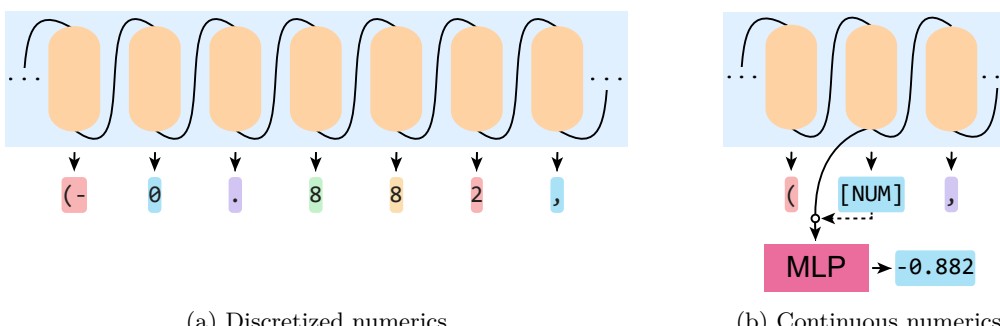

(a) Discretized numerics            (b) Continuous numerics

Figure 2: **Numeric Head.** (Sec. 3.4) Rather than producing digits as discrete tokens (a), we train our model to generate a [NUM] token when a number should be produced. The [NUM] token is used as a mask to signal the embedding should instead be passed through the numeric head, preserving the gradient (b).

**Differences From Traditional Approaches.** The framework presented in this study marks a departure from conventional approaches. Notably, the visual-encoding process does not include graphics-specific inductive biases in its design. It undergoes no training for intermediate vision tasks, such as object detection, segmentation, or attribute regression. The model operates solely on rendered images without access to 3D assets. Moreover, the supervised fine-tuning process employs a training objective not directly related to the physical representation of the scene.

We show in Sec. 4 that these departures do not impede performance or generalization capabilities compared with traditional approaches; in fact, they enhance them. Our experiments demonstrate a compositional-generalization ability without the need for tailor-made designs, surpassing the conventional approach by approximately 90% in OOD shape-recognition accuracy.

### 3.4 Precise Numeric Reasoning in LLMs

Graphics programming requires the precise estimation of continuous quantities such as location and orientation, along with a comprehension of Euclidean space. This requirement extends beyond the coarse, semantic-level spatial reasoning (e.g., "left," "in front of") for which LLMs are typically employed, in tasks such as VQA (Antol et al., 2015). Estimating continuous values through character-based outputs essentially transforms the task into a discrete, combinatorial challenge. In this loss space, prediction errors do not reflect real metric distances – a ground truth value of '4' is considered as close to a prediction of '3' as it is to '8,' highlighting a potential limitation inherent in this approach for learning a continous understanding.

To address this challenge, we introduce a *numeric head* tailored to enable continuous parameter estimation. A visual representation of the numeric-head integration is depicted in Fig. 2b, contrasting with the discrete text-based alternative. The module is implemented as a four-layer MLP that processes the final hidden-layer output of the LLM and transforms it into a scalar value. To allow the LLM to discern between generating numerical values or textual information, we designate a special token in the vocabulary – [NUM] – which serves as a mask to indicate whether a number should be produced. During training, we apply an MSE loss on each number in addition to the next-token prediction loss (Eq. (1)) used on the [NUM] token itself.

We systematically investigate the behavior of character-based and numeric IG-LLM variants for precise spatial reasoning in Sec. 4.2. Our empirical findings support our intuition regarding the limitations of the character-based output and demonstrate that the numeric head enables greater generalization when the testing samples are OOD in parameter space. These differences are highlighted throughout our evaluation.

## 4 Evaluations

To evaluate the ability of our proposed framework to generalize across distribution shifts, we design a number of focused evaluation settings. We conduct experiments on synthetic data in order to quantitatively analyze model capability under controlled shifts.

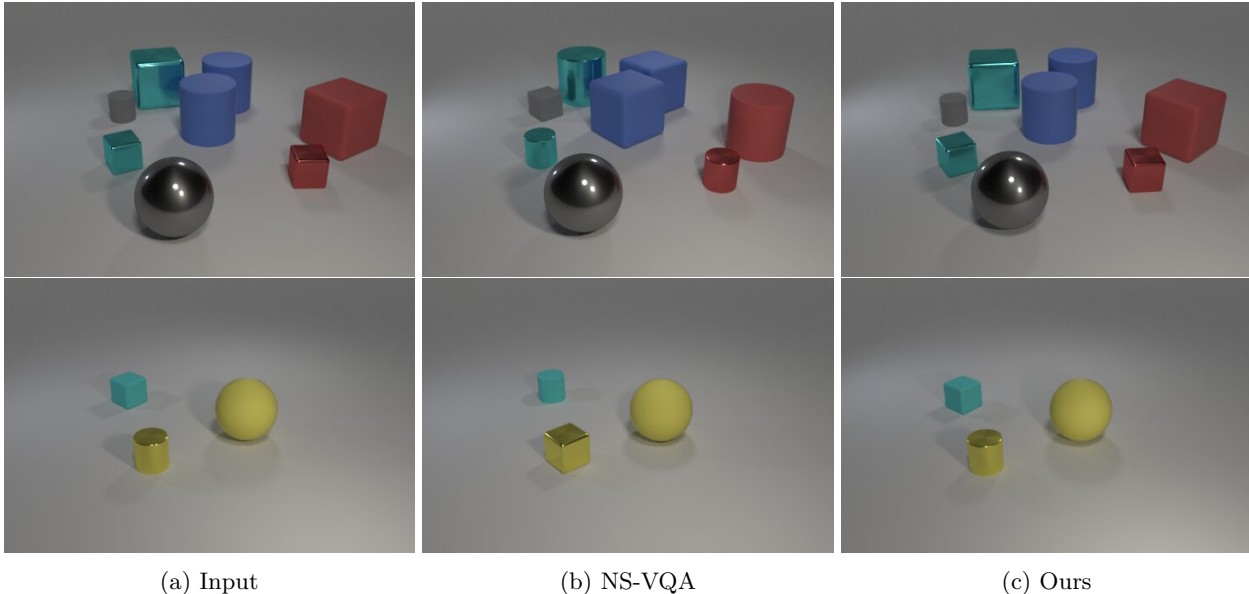

|(a) Input|(b) NS-VQA|(c) Ours|

Figure 3: **OOD CLEVR-CoGenT Samples.** (Sec. 4.1) NS-VQA, with its modular design, fails to disentangle shape from color, while our framework is able to effectively generalize to OOD attribute combinations. See Fig. S.7 for additional samples.

## 4.1 Compositional Generalization on CLEVR

An extension to CLEVR, known as CLEVR-CoGenT (Johnson et al., 2017), serves as a benchmark for evaluating the *compositional*-generalization capabilities of VQA models. This benchmark assesses the model's ability to answer questions about scenes containing objects with unseen combinations of attributes. During training, the dataset is structured such that particular types of objects are only assigned specific combinations of attributes (e.g., blue cubes and red cylinders), while the testing data includes objects with attribute combinations not seen during training (e.g., red cubes and blue cylinders). We adapt this VQA dataset to our inverse-graphics problem domain, employing it for three primary purposes: 1) demonstrating that LLMs can effectively perform inverse graphics by testing on in-distribution (ID) data; 2) illustrating that LLMs exhibit robust compositional generalization to OOD data, while the baseline approach in NS-VQA (Yi et al., 2018) struggles in this setting; and 3) exploring the data-efficiency of our framework.

**Setting.** Following the setting of CLEVR-CoGenT, our training set consists of images of scenes containing objects with only a subset of possible attribute combinations (shape, size, material, and color). In the ID condition, all cubes are rendered in gray, blue, brown, or yellow, and all cylinders are depicted in red, green, purple, or cyan. In contrast, in the OOD condition the color palettes of the shapes are swapped. Spheres are consistently depicted with all eight colors under both conditions. We train both our proposed framework and NS-VQA, our neural-scene de-rendering baseline, on 4k images from the ID condition and evaluate them on 1k images from both the ID and OOD conditions. We follow CLEVR and randomly apply their set of synonyms on the categorical attributes.

**Evaluation Metrics.** To evaluate attribute-recognition accuracy, we employ linear sum assignment on pairwise Euclidean distances to match predicted and ground-truth objects. However, since attribute-recognition accuracy does not account for missing or duplicated objects, we also evaluate the method's ability to produce accurate counts by computing the mean-absolute counting error between the predicted and ground-truth object sets across scenes (*Count*). *L2* represents the positional L2 norm and *Size*, *Color*, *Mat.,* and *Shape* are percentages of attribute-recognition accuracy.

**Results.** Both our proposed framework and NS-VQA achieve >99% accuracy on the ID condition (Tab. 1), which underscores LLMs' ability to perform comparably with domain-specific modular designs. However, when evaluated on the OOD condition, the shape-recognition accuracy of the baseline method drops sub-

Table 1: **CLEVR-CoGenT Results.** (Sec. 4.1) While both our proposed framework and the baseline, NS-VQA, and are able to achieve >99% accuracy on the ID condition, the baseline fails to generalize, with its shape-recognition accuracy dropping by 66.12%. *Color*, *Mat.*, and *Shape* represent respective accuracies and ↑ indicates greater is better.

| | ID | | | OOD | | |
| | Char | Float | NS-VQA | Char | Float | NS-VQA |
|---|---|---|---|---|---|---|
| ↓L2 | 0.21 | 0.16 | 0.18 | 0.22 | 0.17 | 0.18 |
| ↑Size | 99.71 | 99.77 | 100.00 | 99.74 | 99.80 | 100.00 |
| ↑Color | 99.58 | 99.71 | 100.00 | 98.60 | 98.14 | 99.95 |
| ↑Shape | 99.51 | 99.59 | 100.00 | 93.50 | 93.14 | 33.88 |

stantially by 66.12%, while the accuracy of our pipeline decreases by only 6.01%. Notably, when spheres, observed with all colors during training, are removed from the evaluation, the shape accuracy of NS-VQA plummets further to 0.03%. Illustrative reconstructions from the OOD condition can be seen in Fig. 3.

In terms of data efficiency, we find the float-based model to be much more efficient than the char-based model in estimating the positions of objects, but the difference diminishes as the number of samples reaches 4k (Fig. 4). Hypothesizing that the char-based model has learned to compositionally retrieve the exact positions of (individual) objects from the training set rather than learning to effectively interpolate between training values, we measure the likelihood of predicting arbitrary three-decimal values. Our findings reveal that the char-based model is 6.41 times more likely to predict a particular value if that discrete value was observed during training. We further explore float-estimation dynamics in Sec. 4.2.

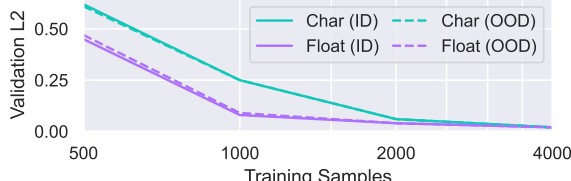

Figure 4: **CLEVR Data Efficiency.** (Sec. 4.1) Plot of the validation L2 positional error by the number of training samples. We observe that the float-based model is consistently more data-efficient but that the difference between the models converges as the number of training samples reaches 4000. See Tab. S.2 for a full quantitative comparison.

## 4.2 Numeric Parameter-Space Generalization

In this section, we investigate the addition of a numeric head and the ability of our framework to generalize across parameter space.

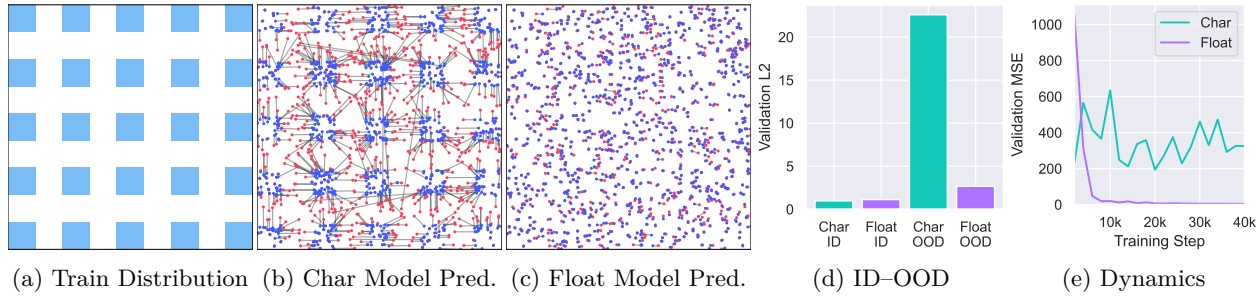

(a) Train Distribution (b) Char Model Pred. (c) Float Model Pred. (d) ID–OOD (e) Dynamics

Figure 5: **2D Parameter-Space Generalization.** (Sec. 4.2.1) (a) Training positions are sampled from the checkerboard. When evaluated on images with uniformly sampled positions, the char-based model fails to generalize outside the training distribution (b) while the float-based model effectively interpolates samples (c). Randomly sampled testing locations are shown in red and the corresponding predictions in blue. (d) shows that, while both methods well estimate samples from the ID condition, the char-based model struggles to generalize. (e) shows a plot of the model's validation MSE as a function of the number of training steps. We observe that the training of the float-based model is much smoother and converges quickly.

### 4.2.1 2D Parameter Space

We begin by testing the framework's ability to generalize in 2D parameter space across range gaps. To accomplish this, we create a dataset comprising 10k images, each featuring a red dot on a white background. During training, the model is shown images where the location of the dot is sampled from a sparse checkerboard grid, as shown in Fig. 5a. During evaluation, the model is shown 1k images where the dot's location is uniformly sampled across the square; points lying outside the checkerboard are effectively OOD inputs.

**Results.** As shown in Fig. 5b, the char-based model exhibits strong overfitting to the training distribution, consistently predicting dot locations restricted to the checkerboard distribution observed during training. In contrast, the float-based model is able to effectively generalize across parameter space, adapting to the uniformly sampled testing distribution during evaluation (Fig. 5c). Although the float-based model exhibits a slight positional bias toward predicting positions on the grid (as evidenced by the higher OOD error), the disparity in the ID–OOD validation-L2 performance gap of the char-based model is 14 times as high as that of the float-based model (Fig. 5d). Moreover, the validation MSE of the float-based model converges quickly to near zero, while the error of the char-based model is much less stable over time (Fig. 5e), suggesting that the float-based model learns smooth, low-dimensional representations of the space while the char-based variant may not.

### 4.2.2 SO(3) Parameter Space

We continue our parameter-space evaluation within the more complex task of SO(3)-pose estimation of orientable objects. For this, we make use of five toy-airplane assets sourced from Super-CLEVR (Li et al., 2023). We construct a training dataset of 10k images of single planes at a fixed location and sampled attributes identical to those in CLEVR. Extending the range-gap setup used in Sec. 4.2.1, the airplanes are assigned random extrinsic-Euler rotations, where the components are sampled from ranges containing inserted gaps (e.g., $[-\frac{\pi}{20}, \frac{\pi}{20}]$). A visual depiction of this space is provided in Fig. 6a, with the training values exclusively sampled from the blue ranges. During testing, we invert the gaps to assess OOD generalization. We evaluate performance across intrinsic-Euler, extrinsic-Euler, axis-angle, and 6D (Zhou et al., 2019) representations.

**Results.** We report full results in Tab. S.3 and here discuss only results from the best-performing representation variants in each evaluation, being intrinsic-Euler for char and 6D for float in ID, and axis-angle for both in OOD. We do so to avoid biasing results with a representation that is better suited for one model variant or the other.

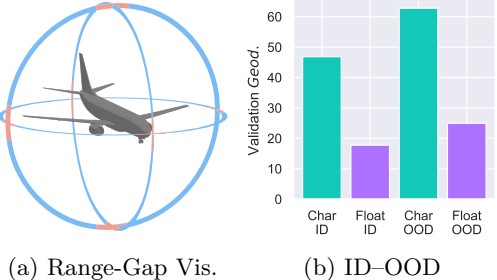

(a) Range-Gap Vis.   (b) ID–OOD

Figure 6: **SO(3) Range Gap.** (Sec. 4.2.2) (a) Visualization of the SO(3) range-gap sampling space. For training, Euler rotations are sampled from the blue regions. In OOD, rotations are exclusively sampled from the red ranges. (b) The float-based model outperforms the char-based variant on validation data sampled from both the ID and OOD conditions. *Geod.* represents geodesic distance in degrees.

As depicted in Fig. 6b, the error of the char-based model is 2.64 times higher than that of the float-based model when evaluated in-distribution. Upon testing in the OOD condition, the disparity is nearly consistent at 2.52 times that observed in the ID scenario, with the ID–OOD gap of the char-based model being 2.21 times that observed in the float-based variant. We attribute the superiority of the float-based model across both conditions to the increased data dimensionality. Additionally, the lesser performance decline observed when evaluating on the OOD gaps further underscores the parameter-space efficiency of the float-based model.

## 4.3 6-DoF Pose Estimation

We examine the ability of our framework to scale in tackling a more challenging inverse-graphics task: that of 6-DoF pose estimation. Our exploration begins with an evaluation on single-object images, encompassing both quantitative and qualitative assessments, where we illustrate the framework's ability to generalize across visual domain shifts. We subsequently extend the setting to include more-complex (albeit, synthetic) multi-

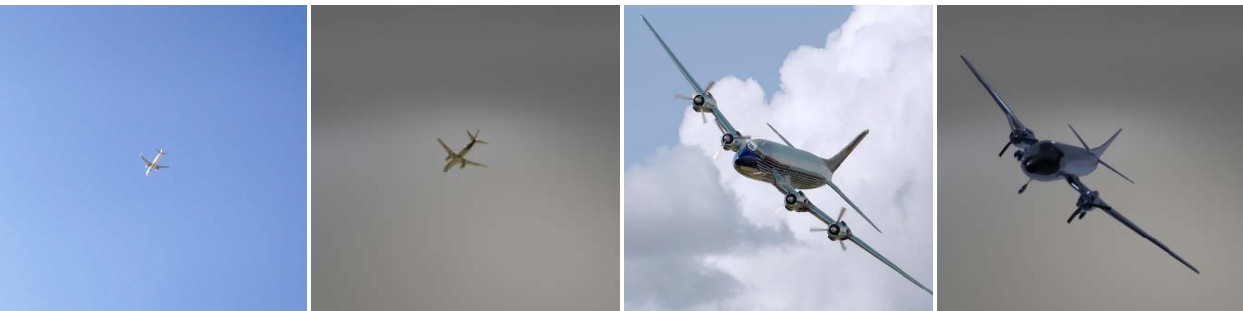

Figure 7: **OOD Single-Object 6-DoF Samples.** (Sec. 4.3.1) A sample 6-DoF reconstruction of real-world images. The model is finetuned with only Blender renderings of toy airplanes that have a white backdrop. See Fig. S.10 for additional samples.

Table 2: **Single-Object 6-DoF Results.** (Sec. 4.3.1) When evaluating on ID data in the one-million-sample single-object 6-DoF eval, we observe little difference between models; both well capture the distribution.

|       | ↓L2  | ↓Geod. | ↑Color | ↑Mat. | ↑Shape |
|-------|------|--------|--------|-------|--------|
| Char  | **0.02** | **5.03** | 79.10 | **99.00** | 99.80 |
| Float | 0.04 | 6.18 | **81.90** | **99.00** | **100.00** |

object scenes, demonstrating promising results for scene estimation, handling larger collections (>100) of diverse assets.

### 4.3.1   Single-Object 6-DoF

We first evaluate our framework's ability to scale to single-object 6-DoF pose estimation. The float- and char-based models are assessed quantitatively using rendered images.

**Setting.** We extend the setting used in Sec. 4.2.2 but unfreeze the previously fixed 3D position and assign it a randomly sampled value. We expand the number of colors used in the dataset to $133^1$ to better emulate the diversity observed in the real world. Differing from the previous setup, we fix the size of the objects due to the relative depth–scale ambiguity of the toy airplanes. To evaluate our framework's ability to scale beyond data-constrained scenarios, we render a training dataset of one-million images. Following the rotation-representation results of Sec. 4.2.2, we use the intrinsic-Euler representation for the char-based model and the 6D representation for the float-based model as their use led to the greatest ID performance.

**Results.** Tab. 2 illustrates that, under this non-data-constrained scenario, both model variants effectively capture the dynamics of the task. The models both notably exhibit a positional error that is an order of magnitude lower than in the CLEVR setting, despite the addition of 3D orientation and an additional positional dimension. They also achieve rotational error that is 28% of that observed in the ID portion of the SO(3) range-gap evaluation. This reinforces the earlier observation from the CLEVR data-efficiency evaluation that, given sufficient data, the model variants exhibit a similar performance ceiling. Still, neither achieves the level of precision necessary to be directly constrained by the three-decimal-place discretization applied to numeric quantities throughout the evaluations nor the 16-bit training precision in the case of the float-based model. See Appendix A for further training details.

As part of our evaluation, we also qualitatively examine the ability of the model to transfer from the renders of toy planes with a solid-white background, on which it was fine-tuned, to estimating the pose and attributes of planes in real-world images. We provide qualitative samples of our model's generalization to such images in Fig. 7. We observe encouraging generalization across the majority of images tested, despite the lack of augmentation or domain-specific inductive bias applied during the training process. However, it

---

[1]https://simple.wikipedia.org/wiki/List_of_Crayola_crayon_colors

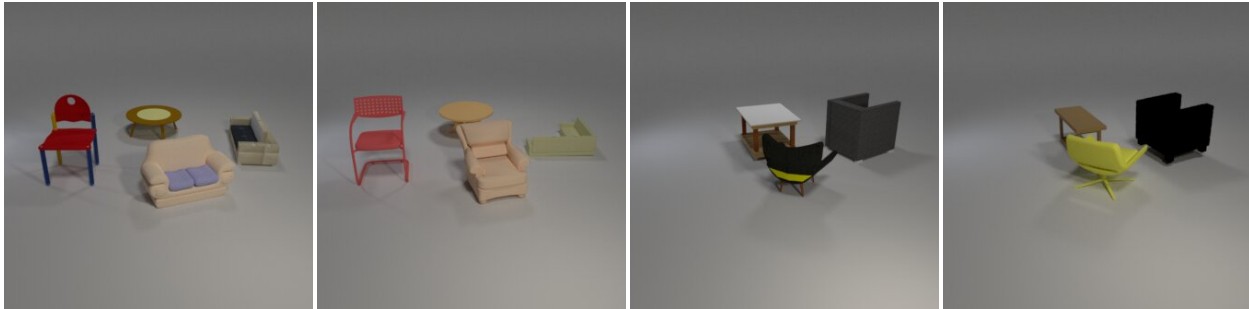

Figure 8: **OOD ShapeNet 6-DoF Samples.** (Sec. 4.3.2) Two sample reconstructions from the OOD ShapeNet 6-DoF pose-estimation experiment. Left to right: input, output. We evaluate on assets not shown during training, with out-of-distribution textures. See Fig. S.9 for additional samples.

is difficult to quantitatively evaluate model performance due to a lack of paired real-world data in line with our compositional task. As a proxy for such an evaluation, we introduce a synthetic setting in Sec. 4.3.2 to quantitatively evaluate the ability of our framework to generalize across visual domains.

### 4.3.2 Scene-Level 6-DoF

In this section, we explore the scalability of our framework to scene-level 6-DoF-pose estimation, featuring 3-5 objects per scene and a much-expanded array of assets. This experiment not only assesses performance under more-challenging conditions but also enables a quantitative evaluation on the framework's ability to generalize to scenes with OOD visual appearances.

**Setting.** We construct an expanded CLEVR-like image–scene dataset, incorporating objects sourced from ShapeNet (Chang et al., 2015). The dataset comprises 56 chair types, 35 sofa types, and 47 table types. We remove the size and material attributes used in CLEVR but employ the expanded color set used in Sec. 4.3.1 to randomly color the objects. After doing so, the total number of possible combinations of attributes is 191-fold that used in the CLEVR-CoGenT experiment. Differing from previous evaluations, we also vary the pitch of the camera and the radius of its arc but maintain a fixed camera focal point. Returning from the million-image single-object 6-DoF evaluation, we render 100k training images and evaluate the framework on three conditions, each with 1K images: (1) ID, which matches the training distribution of scenes with solid-colored objects; (2) OOD texture (OOD-T), where the same object assets are used as in ID but the objects are rendered with original ShapeNet textures instead of randomly assigned solid colors; and (3) OOD encompassing both unseen objects and original ShapeNet textures (OOD-T+S). We use this to emulate the distribution shift of modeling real-world scenes while facilitating quantitative evaluation.

Table 3: **ShapeNet 6-DoF Results.** (Sec. 4.3.2) The float-based model outperforms the char-based variant across all evaluations. *Chamf.* represents the Chamfer distance between the ground-truth and estimated scenes. *Cat.* represents category accuracy (sofa, chair, table).

|  | ID | | OOD-T | | OOD-T+S | |
|---|---|---|---|---|---|---|
|  | Char | Float | Char | Float | Char | Float |
| ↓L2 | 0.23 | **0.21** | 0.28 | **0.26** | 0.37 | **0.36** |
| ↓Geod. | 8.05 | **5.78** | 12.95 | **9.79** | 44.47 | **41.31** |
| ↓Count | **0.01** | **0.01** | **0.04** | 0.05 | **0.05** | **0.05** |
| ↑Color | 80.96 | **84.27** | N/A | N/A | N/A | N/A |
| ↑Shape | 91.96 | **94.05** | 76.89 | **81.67** | N/A | N/A |
| ↑Cat. | 97.92 | **98.58** | 96.48 | **97.65** | 86.52 | **88.23** |
| ↓Chamf. | 0.41 | **0.24** | 0.72 | **0.50** | 2.56 | **2.44** |

**Results.** We observe that both approaches scale to the task, though the float-based model outperforms – or ties with – the char-based variant across evaluations (Tab. 3).

There is a decrease in performance observed when stepping to the OOD-T setting, which is most-strongly observed in the count error (x4–5) and the shape-recognition accuracy (-15.07% in char and -12.38% in float). We empirically attribute this to the model occasionally explaining some multi-color textured objects using a composition of multiple, solid-color assets. Quantitatively supporting this, the performance decrease is not as strongly reflected in scene-level chamfer distance (x1.8–2.1).

See Fig. 8 for sample reconstructions from OOD-T+S and Fig. S.8 for samples from the ID setting. We additionally test our model on real-world samples but find that it fails to consistently generalize (Fig. S.1). We attribute this failure as likely due primarily to limitations in the training camera-position distribution.

## 5 Discussion and Limitations

Through our investigation, we demonstrated the ability of LLMs to facilitate inverse-graphics tasks across a variety of domain shifts, albeit within controlled settings. In designing targeted evaluations to analyze the model's generalization ability, our goal was to lay the groundwork necessary for future advancements. However, scaling up these models to metrically reconstruct complex real-world scenes will undoubtedly pose additional challenges.

The primary limitation of our approach lies in that its expressiveness is constrained by the expressiveness of the training-data-generation framework. We demonstrated its ability to learn to compositionally disentangle images of scenes into constituent elements, reconstructing scenes under distribution shifts. However, reproducing scenes as text, it can reconstruct scenes containing unknown objects in OOD configurations, but it does so in terms of the objects – and language – it is trained with. If it does not know the name of asset `chairs_0055`, it will not be able to use it. Even if the model produces the name of a new color or shape from outside the training data, the graphics engine rendering the LLM output must have an understanding of it in order to apply it.

In contrast, the generality of our approach, which doesn't incorporate special task-specific inductive biases, allows it to scale with the diversity of the training data or the expressivity of the code format, beyond simple shapes. Future work may explore more-scalable training-data generators or integrate self-supervision techniques to enable learning from unlabeled images. In order to better focus our investigation on generalization across distributional shifts, we employ a relatively straightforward object-centric code representation, though more-expressive scene representations should also be explored.

Our evaluation scenes feature only minor object occlusions and are relatively simple. While a generic next-token-prediction objective paired with MSE float supervision sufficed for these scenarios, addressing harder-to-disentangle scenes may require a trade-off between generality and inductive bias to incorporate additional supervision such as differentiable rendering.

## 6 Conclusion

In this work, we investigated the ability of LLMs to solve inverse-graphics challenges. Introducing the Inverse-Graphics Large Language Model (IG-LLM) framework, we demonstrated that the broad generalization and reasoning capabilities of LLMs can be harnessed to facilitate inverse-graphics tasks. Through extensive evaluation, we assessed the model's capacity to generalize out-of-domain, revealing its ability to abstract scene elements compositionally. We additionally explored the integration of a numeric head to adapt LLMs for continuous metric-value estimation, providing enhanced generalization and smoother training dynamics. Our quantitative analyses demonstrate its ability to generalize compositionally (Sec. 4.1), in parameter space (Sec. 4.2), and across visual domains (Sec. 4.3). Our investigation demonstrates the ability of IG-LLM to leverage the general knowledge of LLMs in solving inverse-graphics problems, opening a new avenue for research.

**Acknowledgements**   We thank Silvia Zuffi for useful discussions and Benjamin Pellkofer for IT support.

**Disclosure**   MJB has received research gift funds from Adobe, Intel, Nvidia, Meta, and Amazon. MJB has financial interests in Amazon, Datagen Technologies, and Meshcapade GmbH. While MJB is a consultant for Meshcapade, his research in this project was performed solely at, and funded solely by, the Max Planck Society.

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

## A    Further Training Details

We finetune the LLaMA 1-based Vicuna 1.3 model[2] with LoRA (Hu et al., 2022a). We use the HuggingFace Transformers and PEFT libraries, along with DeepSpeed `ZeRO-2` (Rajbhandari et al., 2020). In all experiments, we use a `lora_r` of 128, a `lora_alpha` of 256, a LoRA learning rate of `2e-05`, a linear projector learning rate of `2e-05`, a numeric head learning rate of `2e-04`, and a cosine learning-rate schedule. All models are trained with an effective batch size of 32 with `bfloat16` mixed-precision training. Both the cross-entropy next-token-prediction and mean-square-error (MSE) losses are given a weight of 1.

The models for the CLEVR and parameter-space generalization experiments are trained for `40k` steps. The single-object 6-DoF pose-estimation model is trained for `200k` and the scene-level ShapeNet model for `500k` steps.

We use the frozen CLIP visual tokenizer from [3]. This CLIP variant has an input size of 336x336 pixels. For the CLEVR evaluation, we render images at the original size of 480x320 to ensure compatibility with NS-VQA, but we pad and resize them for use with our model. For the remaining evaluations, we directly render images at a resolution of 336x336.

We employ greedy token sampling across evaluations.

## B    Further CLEVR Data-Generation Details

The original CLEVR dataset is rendered with random positional jitter in both the lights and camera. This information is not recorded in the public dataset, so we re-render CLEVR-CoGenT with a fixed camera position, but maintain the randomness in the lighting.

## C    Further Numeric-Head Details

Our numeric head is composed of a `tanh` layer, followed by a linear layer, a `GELU` activation, and a final linear projection. The final LLaMA hidden state is passed through an RMS norm (Zhang & Sennrich, 2019) before it is shared between the token head and numeric head, which rescales but does not recenter the embedding.

During training, the locations of these tokens in the ground-truth sequence are known so they can be masked to apply the MSE loss. During sampling, the position of these tokens is not pre-known and is dependent on the generated sequence. We first generate the token-only sequence and then substitute the estimated numbers back in a single additional pass.

## D    ShapeNet 6-DoF YOLOX-6D-Pose

While our investigation focused on the ability of our framework to generalize across distribution shifts, and our intention was not to suggest it as optimal over task-specific approaches to these particular problems, we additionally train and evaluate YOLOX-6D-Pose (Maji et al., 2024) on our ShapeNet scene-level 6-DoF setting to serve as a quantitative point of comparison against a recent state-of-the-art specialized method. YOLOX-6D-Pose is a modular approach trained to detect objects and regress their 6-DoF pose. In contrast to our framework, which learns to disentangle images into compositional scene representations solely by modeling the token sequence, YOLOX-6D-Pose relies on bounding-box supervision, known camera parameters, access to 3D assets, hand-crafted losses, and extensive task-specific data augmentation. By foregoing such inductive biases in the training of our framework, we sought to keep it deliberately generic to maintain the focus of our investigation on generalization without relying on task-specific designs, though such supervision and data augmentation could also be applied to our approach.

Following Maji et al. (2024), we train the extra-large YOLOX-6D-Pose variant for 300 epochs, enabling all components. However, the model cannot be compared directly to our framework. Because the detector

---

[2]https://huggingface.co/lmsys/vicuna-7b-v1.3
[3]https://huggingface.co/openai/clip-vit-large-patch14-336

scores object localizations with a confidence rather than producing a definite object set, we select the top-n scoring detections equal to the ground-truth number of objects in the scene after non-maximum suppression is performed. The method is additionally unable to compositionally estimate object attributes (i.e., color). The goal of the OOD ShapeNet evaluation was to examine the ability of our framework to generalize across visual domains with OOD texture and shape serving as a quantifiable proxy for real-world visual generalization. In contrast to YOLOX-6D-Pose, which is trained to be color-invariant through visual augmentation, our method, tasked to estimate color, must be color-*variant*. This complicates the evaluation of generalization ability.

Evaluation metrics are shown in Tab. S.1. YOLOX-6D-Pose excels in-distribution, estimating 6D rotations to, on average, one degree of error. However, despite its strong augmentation, this performance does not transfer out-of-distribution, with its scene-level chamfer distance increasing 19-fold in the OOD-T setting as opposed to the 2x increase in the float-based IG-LLM variant, which also outperforms it. This suggests that extensive data augmentation alone may not be enough to ensure out-of-distribution generalization.

Table S.1: **ShapeNet 6-DoF YOLOX-6D-Pose Results.** (Appendix D) We observe that while YOLOX-6D-Pose (Maji et al., 2024) excels in the ShapeNet setting in-distribution, it does not as consistently generalize.

| | ID | | OOD-T | | OOD-T+S | |
|---|---|---|---|---|---|---|
| | Maji et al.* | Float | Maji et al.* | Float | Maji et al.* | Float |
| ↓L2 | 0.06 | 0.21 | 0.19 | 0.26 | 0.96 | 0.36 |
| ↓Geod. | 1.20 | 5.78 | 6.02 | 9.79 | 41.62 | 41.31 |
| ↓Count | N/A | 0.01 | N/A | 0.05 | N/A | 0.05 |
| ↑Color | N/A | 84.27 | N/A | N/A | N/A | N/A |
| ↑Shape | 98.16 | 94.05 | 90.53 | 81.67 | N/A | N/A |
| ↑Cat. | 99.18 | 98.58 | 96.96 | 97.65 | 80.47 | 88.23 |
| ↓Chamf. | 0.04 | 0.24 | 0.77 | 0.50 | 3.74 | 2.44 |

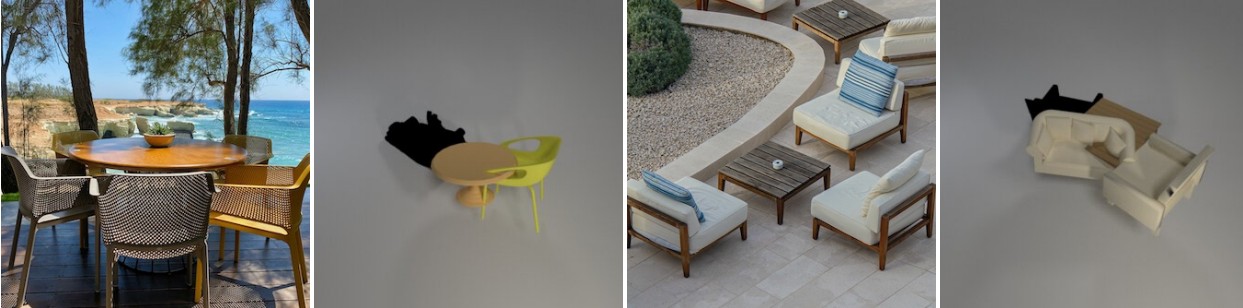

Figure S.1: **Real-World ShapeNet 6-DoF Samples.** (Sec. 4.3.2) Real-world sample reconstructions from the ShapeNet 6-DoF pose-estimation experiment. We observe that the model is sensitive to OOD camera configurations. During data generation, the camera is assigned a random pitch and radius, with its optical axis fixed passing through the global origin. As such, we find that the model learns the bias and is limited by the expressivity of the training-data-generation framework, and, while it effectively interpolates values, it struggles to extrapolate outside of the camera configurations on which it was trained on. We observe that the model is still, however, often able to identify the first few most-salient objects in the scene and produce meaningful assets (the first two in each of these samples being the rightmost chair then the table) before attempting to explain background features.

Table S.2: **Full CLEVR Data-Efficiency Results.** (Sec. 4.1)

(a) ID

| | ↓L2 | ↓Count | ↑Size | ↑Color | ↑Mat. | ↑Shape |
|---|---|---|---|---|---|---|
| **500** | | | | | | |
| Char | 1.15 | **0.30** | 87.58 | 78.23 | 87.09 | 83.25 |
| Float | **0.98** | 0.44 | **91.43** | **85.51** | **90.73** | **89.29** |
| **1000** | | | | | | |
| Char | 0.73 | **0.18** | 97.14 | 94.54 | 96.50 | 95.98 |
| Float | **0.39** | 0.18 | **98.96** | **98.69** | **98.53** | **98.40** |
| **2000** | | | | | | |
| Char | 0.35 | **0.08** | 99.57 | **99.35** | **99.09** | **99.30** |
| Float | **0.26** | 0.09 | 99.55 | 99.28 | 99.04 | 99.23 |
| **4000** | | | | | | |
| Char | 0.21 | **0.05** | 99.71 | 99.58 | 99.27 | 99.51 |
| Float | **0.16** | **0.05** | **99.77** | **99.71** | **99.53** | **99.59** |

(b) OOD

| | ↓L2 | ↓Count | ↑Size | ↑Color | ↑Mat. | ↑Shape |
|---|---|---|---|---|---|---|
| **500** | | | | | | |
| Char | 1.13 | **0.36** | 87.21 | 75.51 | 85.57 | 79.50 |
| Float | **1.01** | 0.53 | **90.71** | **79.45** | **89.24** | **84.42** |
| **1000** | | | | | | |
| Char | 0.74 | **0.21** | 96.25 | 92.19 | 94.87 | 90.45 |
| Float | **0.41** | 0.23 | **98.92** | **96.49** | **97.75** | **94.75** |
| **2000** | | | | | | |
| Char | 0.36 | **0.11** | 99.52 | **97.56** | 98.66 | 92.26 |
| Float | **0.28** | 0.12 | 99.29 | 97.33 | **98.66** | **94.76** |
| **4000** | | | | | | |
| Char | 0.22 | 0.06 | 99.74 | **98.60** | **99.33** | **93.50** |
| Float | **0.17** | **0.05** | **99.80** | 98.14 | 99.21 | 93.14 |

Table S.3: **Full SO(3) Range-Gap Results.** (Sec. 4.2.2)

(a) ID

| | ↓Geod. | ↑Size | ↑Color | ↑Mat. | ↑Shape |
|---|---|---|---|---|---|
| **Char** | | | | | |
| Ext-Euler | 67.31 | 99.80 | 100.00 | 97.80 | 98.70 |
| Int-Euler | **46.86** | 99.90 | 100.00 | 97.10 | 98.30 |
| AA | 53.74 | 100.00 | 100.00 | 97.40 | 98.30 |
| 6D | 77.69 | 100.00 | 99.90 | 97.40 | 98.60 |
| **Float** | | | | | |
| Ext-Euler | 41.25 | 100.00 | 100.00 | 98.00 | 98.60 |
| Int-Euler | 27.05 | 99.90 | 100.00 | 97.70 | 99.30 |
| AA | 26.58 | 100.00 | 100.00 | 97.50 | 99.00 |
| 6D | **17.76** | 100.00 | 100.00 | 97.40 | 99.30 |

(b) OOD

| | ↓Geod. | ↑Size | ↑Color | ↑Mat. | ↑Shape |
|---|---|---|---|---|---|
| **Char** | | | | | |
| Ext-Euler | 78.21 | 100.00 | 99.90 | 97.50 | 99.40 |
| Int-Euler | 68.50 | 100.00 | 100.00 | 97.90 | 99.30 |
| AA | **62.80** | 100.00 | 100.00 | 98.10 | 99.30 |
| 6D | 104.53 | 100.00 | 99.90 | 97.20 | 99.00 |
| **Float** | | | | | |
| Ext-Euler | 42.03 | 100.00 | 100.00 | 97.20 | 99.10 |
| Int-Euler | 43.49 | 100.00 | 100.00 | 97.30 | 99.40 |
| AA | **24.96** | 100.00 | 100.00 | 97.80 | 99.40 |
| 6D | 27.12 | 100.00 | 100.00 | 98.10 | 99.50 |

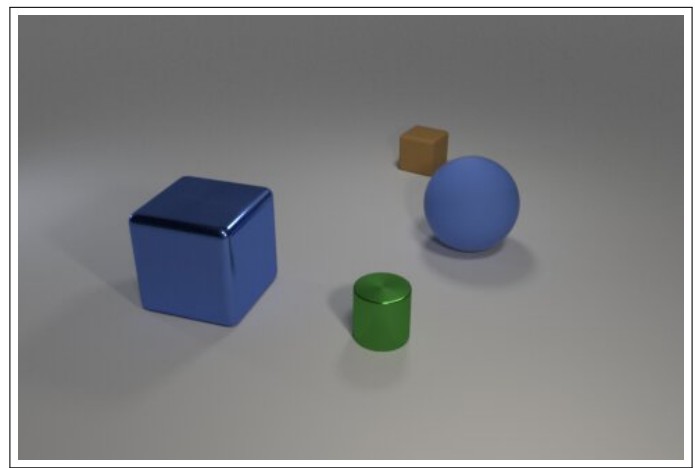

```
add(color='green', size='tiny', material='shiny', shape='cylinder', loc=(2.163, -1.384,
↪  0.350))
add(material='metal', rotation=-0.126, shape='cube', loc=(-0.033, -2.456, 0.700),
↪  color='blue', size='large')
add(size='large', material='rubber', color='blue', loc=(1.352, 1.165, 0.700),
↪  shape='sphere')
add(color='brown', material='matte', shape='cube', size='tiny', loc=(-1.185, 2.816,
↪  0.350), rotation=0.144)
```

Figure S.2: **CLEVR-CoGenT Train Sample.** (Sec. 4.1)

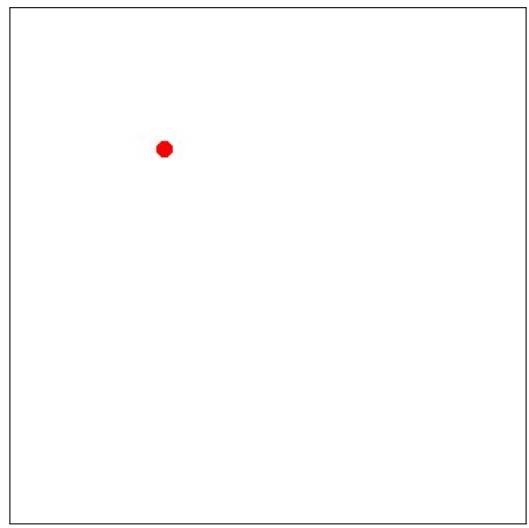

```
add(x=0.292, y=0.266)
```

Figure S.3: **2D Parameter-Space-Generalization Train Sample.** (Sec. 4.2.1)

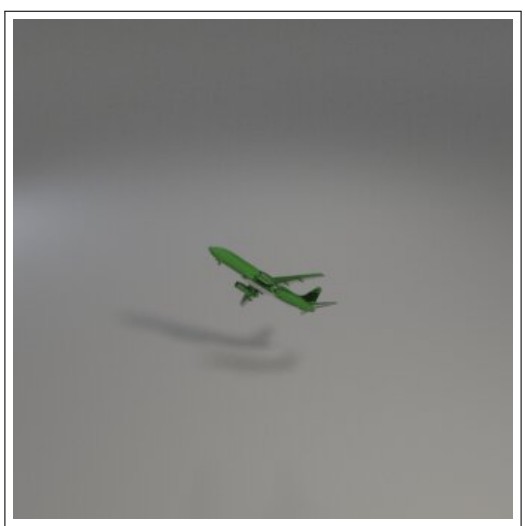

```
add(shape='airliner', size='tiny', color='green', material='matte', rotation=(-0.798,
↪   0.124, 0.590, -0.562, -0.507, -0.654))
```

Figure S.4: **SO(3) Range-Gap Train Sample.** (Sec. 4.2.2)

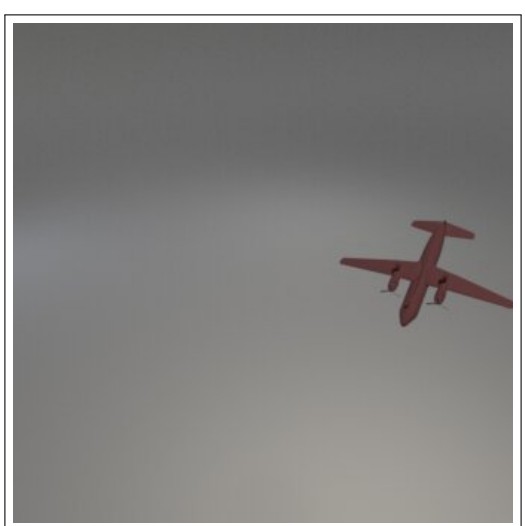

```
add(loc=(6.355, -4.600, 4.206), color='Mahogany', shape='jet', material='matte',
↪   rotation=(0.941, -0.337, 0.022, -0.303, -0.815, 0.493))
```

Figure S.5: **Single-Object 6-DoF Train Sample.** (Sec. 4.3.1)

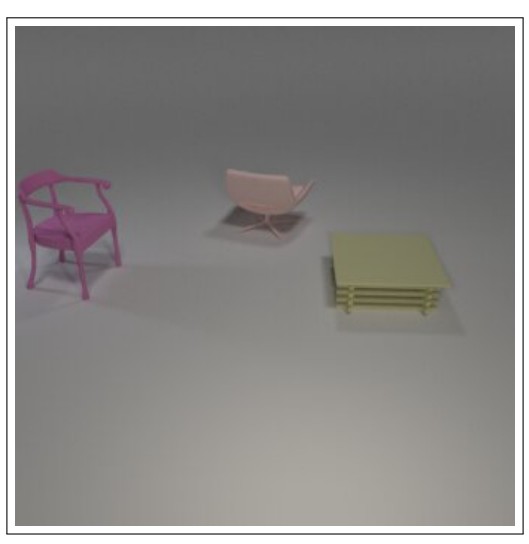

```
add(rotation=(0.999, 0.024, -0.042, 0.024, 0.496, 0.868), color='Olive Green',
↪   loc=(2.228, 0.057, -10.362), shape='tables_0010')
add(rotation=(0.934, 0.177, -0.310, 0.163, 0.561, 0.812), loc=(0.032, 1.816, -12.639),
↪   color='Melon', shape='chairs_0055')
add(loc=(-3.707, 1.009, -10.332), rotation=(-0.235, 0.481, -0.845, -0.689, -0.696,
↪   -0.204), shape='chairs_0008', color='Red Violet')
```

Figure S.6: **ShapeNet 6-DoF Train Sample.** (Sec. 4.3.2)

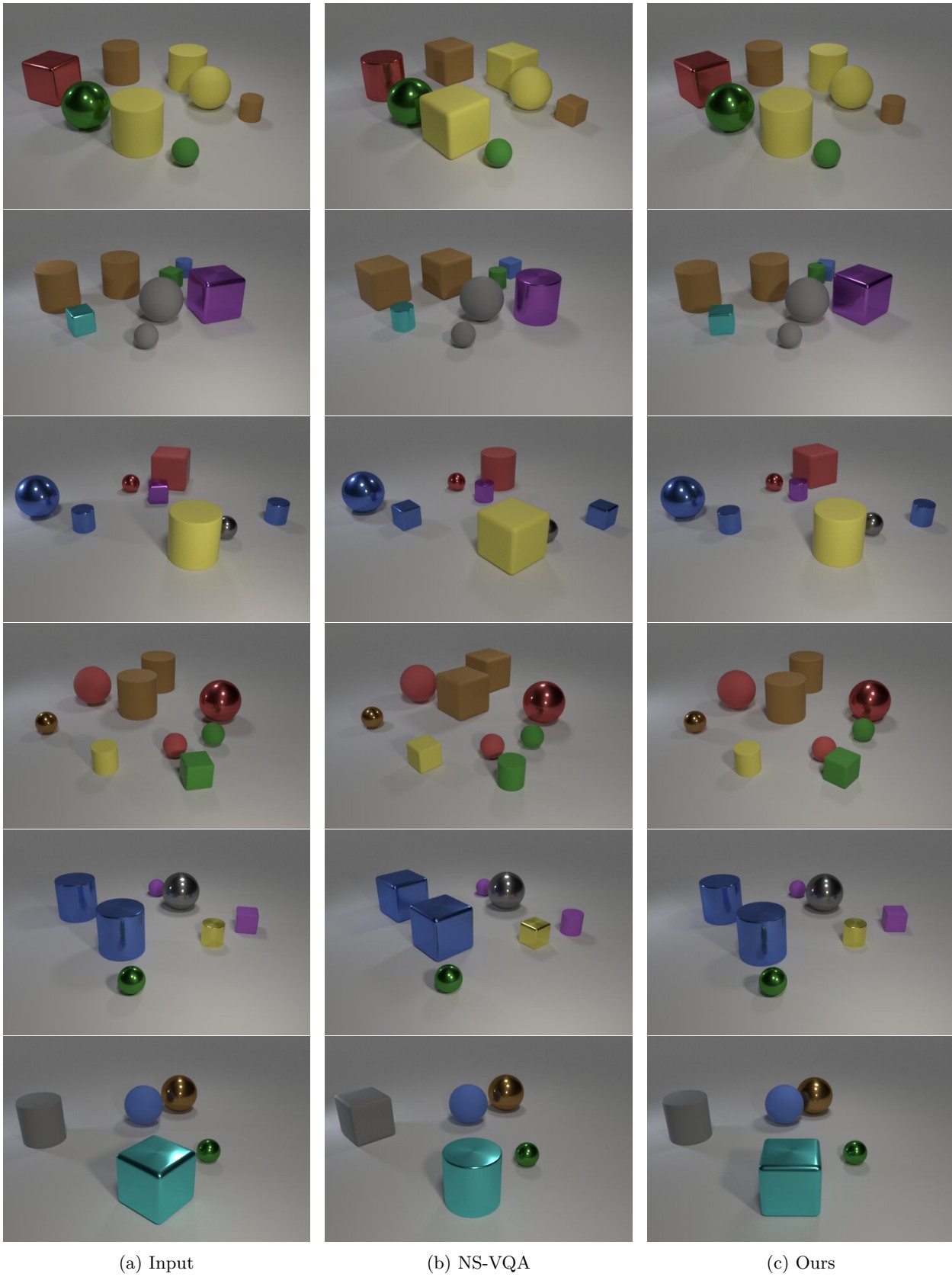

(a) Input        (b) NS-VQA        (c) Ours

Figure S.7: **Additional OOD CLEVR-CoGenT Samples.** (Sec. 4.1)

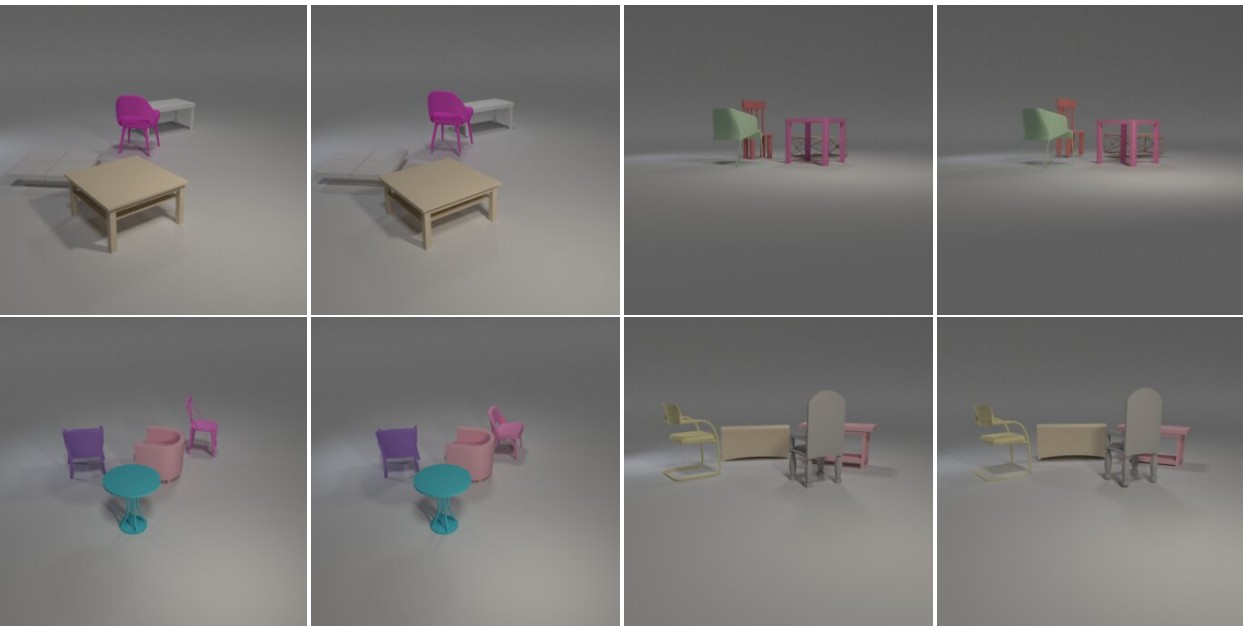

Figure S.8: **ID ShapeNet 6-DoF Samples.** (Sec. 4.3.2) Input–output pairs are shown left-to-right.

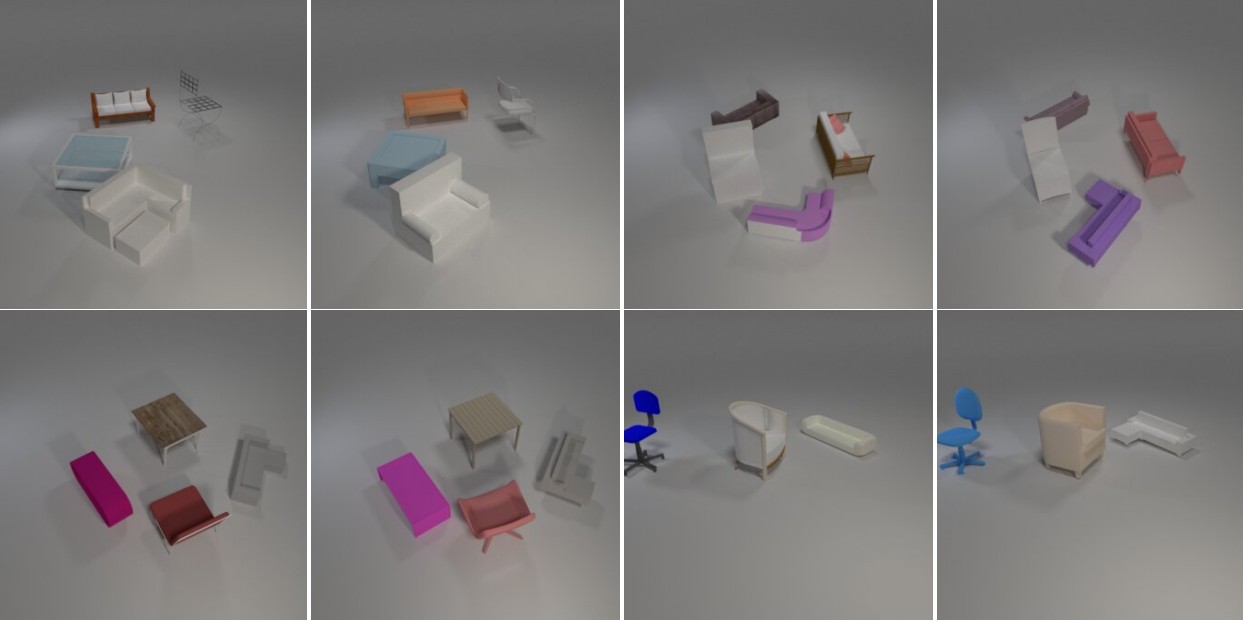

Figure S.9: **Additional OOD ShapeNet 6-DoF Samples.** (Sec. 4.3.2) Input–output pairs are shown left-to-right.

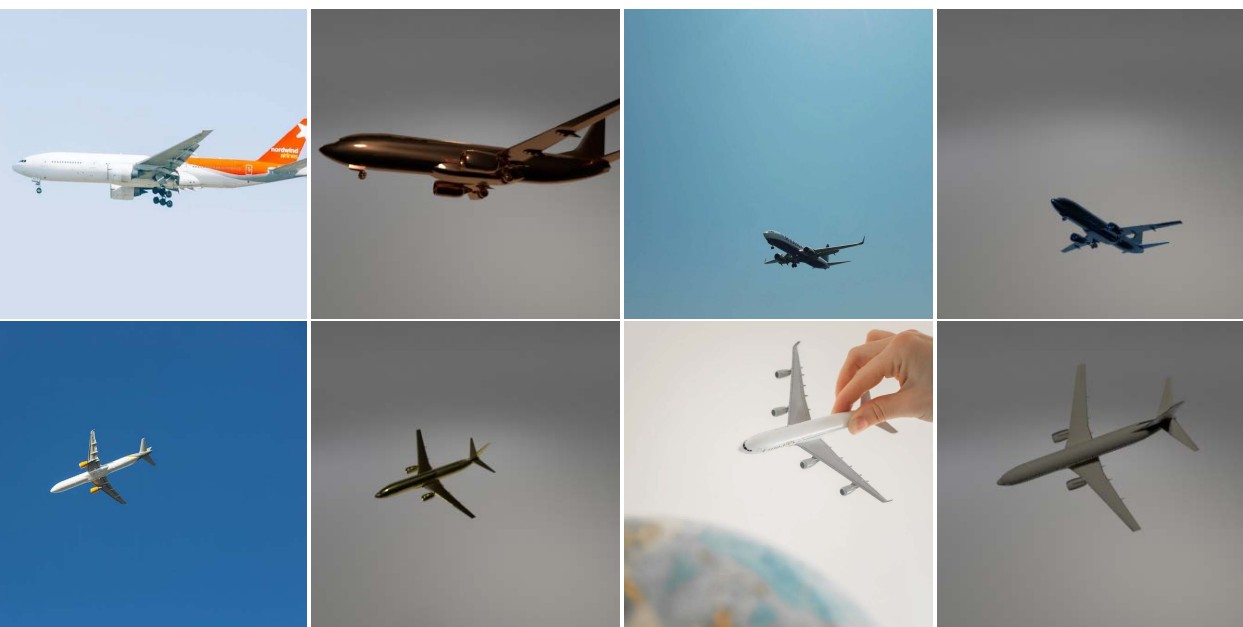

Figure S.10: **Additional OOD Single-Object 6-DoF Samples.** (Sec. 4.3.1) Input–output pairs are shown left-to-right.

