# OpenReview forum: "Re-Thinking Inverse Graphics With Large Language Models"
_TMLR — Accepted by TMLR_

### Review · Reviewer_ziKs · 2024-05-19

**Summary Of Contributions:**

This paper proposes a method for inferring the rendering code that can be used in a rendering engine to reproduce a given image. The authors build a synthetic dataset to fine-tune the pre-trained multi-modality LLM, and relevant experiments have been conducted to analyze the capabilities of the finetuned model. To the best of my knowledge, this work is the first to utilize LLMs in inverse graphics tasks.

**Audience:**

Yes

**Claims And Evidence:**

Yes

**Requested Changes:**

It is recommended that the author include a comparison with existing methods in object pose estimation to more intuitively show the performance level of the method. This method can predict the color and material of objects, but these properties were not involved in the out-of-distribution experiments. In theory, CLIP will extract this information, but it is unclear whether LLMs can utilize it. The author has summarized the limitations of the method, and it would be better to visualize the results caused by these limitations.

**Strengths And Weaknesses:**

This paper presents an effective approach to a type of inverse graphics problem. It utilizes the powerful scene interpretation capabilities of multi-modal LLMs to infer the properties of objects, offering a new perspective on this task. The authors conducted experiments to demonstrate the model's strong performance on out-of-distribution data.

However, the approach requires the availability of object templates, and the object shapes are very simple, which limits its practicality for real-world applications. The authors conducted experiments only with their method. In some tasks, such as object pose estimation, their method could be compared with existing methods, but they did not perform such comparisons.

---

> ### Author Response · Authors · 2024-07-02
> **Response by Authors**
>
> We thank the reviewer for their thoughtful comments and suggestions, which will help to improve the clarity of our paper. We address individual points below:
> ## Object Templates
> We apologize, but we are not clear about the reviewer's comment regarding object templates. We respond to two possible interpretations:
> 1. Use of 3D Assets in Constructing Datasets. If the reviewer is referring to the use of 3D assets in constructing the images of the training and evaluation datasets, we agree there are limitations in the expressivity of static assets. We employed a relatively straightforward object-centric code representation across experiments to evaluate our framework's ability to generalize across distribution shifts. Following the focus of our investigation on generalization, we agree that exploration of more-expressive scene representations will be interesting. We have noted this in the future-work section, and we have reworded this portion to improve clarity.
> 2. Use of Templates in Supervision. If the comment is about the use of templates as supervision, we'd like to clarify that our approach does not incorporate object templates during training or inference. Training is performed via a standard next-token-prediction cross-entropy loss and, in the float-based model variant, an MSE loss on estimated quantities. Our model learns to disentangle images into compositional-scene representations solely by modeling the text sequence, without incorporating differentiable rendering, segmentation maps, or bounding boxes. We have added a phrase to better clarify this.
> ## Shape Complexity
> We agree with the reviewer that the shapes used in our evaluation are relatively simple. However, our approach is agnostic of object representation and does not have access to the underlying 3D assets, so the shapes could be simply replaced with other assets.
>
> We employ the standard CLEVR-CoGenT dataset to evaluate the ability of our approach to generalize compositionality. While this dataset uses only three shape primitives, we later demonstrate our approach's scalability by reconstructing scenes with >100 unique ShapeNet objects.
> ## Comparisons
> We thank the reviewer for their suggestion of including additional comparisons. Representative of the traditional neural-scene de-rendering paradigm, we train and evaluate NS-VQA's de-rendering framework on the CLEVR-CoGenT split in Section 4.1. Our intention was not to claim our approach as optimal for these particular tasks -- which may be better solved using a task-specific approach -- but to investigate the ability of our framework to generalize across focused distribution shifts, where relative performance differences can be observed. Regarding the evaluations of Section 4.3.2, we are not aware of works that are directly comparable, but we will adapt a baseline approach such as [1] to be incorporated.
> ## OOD Color
> Color is explicitly part of the CLEVR-CoGenT evaluation in Section 4.1. During training all cubes are gray, blue, brown, or yellow and all cylinders are red, green, purple, or cyan. These compositional pairings are swapped during the OOD evaluation to test compositional-generalization ability. We also quantitatively evaluate generalization across visual-distribution shifts (asset texture) in Section 4.3.2.
> ## Failure-Case Visuals
> We include failure-case visualizations in supplementary Figure S.1.
>
> [1] Maji, Debapriya, et al. "YOLO-6D-Pose: Enhancing YOLO for Single-Stage Monocular Multi-Object 6D Pose Estimation." 2024 International Conference on 3D Vision (3DV). IEEE, 2024.

---

> > ### Author Response · Authors · 2024-07-11
> > **Response by Authors**
> >
> > As suggested by the reviewer, we have included a comparison in the ShapeNet scene-level 6-DoF evaluation in Appendix D. We thank the reviewer for their comments and suggestions, and we would be happy to answer any remaining questions.

---

### Review · Reviewer_wDPr · 2024-06-07

**Summary Of Contributions:**

The paper propose to inverse graphics by exploiting LLM with image embedding from pre-trained models. It is an interesting attempt to explore the capabilities of LLMs in the task of understanding scene geometry.

**Audience:**

Yes

**Claims And Evidence:**

Yes

**Requested Changes:**

Please refer to Strengths And Weaknesses.

**Strengths And Weaknesses:**

### Strengths
* The proposed method is novel, which utilizes LLM to understand the spatial information embedded in images. It reveals the potential to employ LLM to manipulate objects in graphic engines.
* The selected synthetic datasets are intuitive and efficient to support the claims.

### Weaknesses
* Some metrics for evaluation are not clearly defined, such as Tab. 2.
* Most objects in the dataset are synthetic and symmetric, with a clean background. They are too simple compared with real scenes.
* It seems that no other existing methods are compared quantitatively with the proposed approach, in Section 4.2 & 4.3.

---

> ### Author Response · Authors · 2024-07-02
> **Response by Authors**
>
> We thank the reviewer for their comments, and we appreciate that they find our evaluations interesting, intuitive, and supportive of our claims.
> ## Metric Definitions
> In response to the reviewer's suggestion, we have revised the paper to provide a more-detailed explanation of each metric upon its first reference.
> ## Shape Complexity
> We agree with the reviewer that the shapes used in our evaluations are simpler than those found in real scenes. We use synthetic data to assess generalization across controlled distribution shifts, building a foundation for future work. We acknowledge that scaling to metrical reconstruction of natural scenes will undoubtedly pose additional challenges. In Section 3.3, we characterize the shapes in the CLEVR setting as "simple," and we have also added a phrase to the limitations section.
> ## Comparisons
> The purpose of Sections 4.2.1 and 4.2.2 -- and to some extent Section 4.3.1 -- was to investigate the dynamics of inverse-graphics continuous-parameter estimation in LLMs. The tasks used, such as locating a red dot on a white background, serve to investigate generalization but are not intended as generally applicable benchmarks for evaluating task-specific approaches. For instance, the dot problem could be mastered using a simple `argmin`. The evaluation in Section 4.2.2 may be applied to highlight the limitations of discrete vs. continuous representations, such as [1], which classifies pose into discrete bins.
>
> Similarly, Section 4.3.2 was designed to evaluate our framework's ability to generalize across visual-domain shifts. Our intention was not to propose our approach as optimal for that particular task, which may be better solved using a task-specific model. Given that, we are not aware of directly comparable works, but we will adapt a baseline approach such as [2] to be incorporated into the evaluations of Section 4.3.2.
>
> [1] Kehl, Wadim, et al. "SSD-6D: Making RGB-Based 3D Detection and 6D Pose Estimation Great Again." Proceedings of the IEEE International Conference on Computer Vision. 2017.
>
> [2] Maji, Debapriya, et al. "YOLO-6D-Pose: Enhancing YOLO for Single-Stage Monocular Multi-Object 6D Pose Estimation." 2024 International Conference on 3D Vision (3DV). IEEE, 2024.

---

> > ### Author Response · Authors · 2024-07-11
> > **Response by Authors**
> >
> > As the reviewer suggested, we have added a comparison to the ShapeNet scene-level 6-DoF evaluation to Appendix D. We would be happy to answer any remaining questions. We thank the reviewer for their time and thoughtful evaluation.

---

### Review · Reviewer_BjuB · 2024-06-18

**Summary Of Contributions:**

Large language models (LLMs) / Large Multimodal Models (LMMs) have exhibited impressive performance in solving language and vision tasks. Inverse graphics is a challenging task that invert images into physical variables to enable reproduction of the observed scene. In this paper, authors explores how to harness the powers of LLMs / LMMs to decode visual embeddings into a structured and compositional 3D-scene representation. Experimental results also demonstrate the potiental of using LLMs in solving inverse Graphics tasks.

**Audience:**

Yes

**Broader Impact Concerns:**

This paper does not have any borader impact concerns.

**Claims And Evidence:**

Yes

**Requested Changes:**

1. The related works should be improved. The current works just mention a lot of works and do not well introduce the background about inverse graphics (e.g., why we need to do it and how we do it), and also lack many works about LLMs.


**Minor issues**
1. This paper mentioned many LLM usages. However, LLMs are usually used to process language-only tasks (e.g., GPT-1, GPT-2, GPT-3 and ChatGPT-3.5).  In this paper, To be more precise, the models used in this paper is Large Vision-Language Models (LVLMs) or Large Multi-modal Models (LMMs). I think authors should acknowledge this point.

**Strengths And Weaknesses:**

**Strengths**
This paper has studied how to harness the capability of LLMs to solve inverse graphics tasks.

**Weaknesses**
1. The design of the proposed architectures is simple and lack novelty, which just follows original vision-language models, that use generated 3D data and instructions to train the corresponding IG-LLMs.
2. In Table 1, it seems the used datasets have achieved nearly 99% precision. Is it really challenging for this task or demonstrate some over-fitting issues? I think more challenging tasks should be provided to prove the generalization of the proposed method.
3. Do you try some other LLMs as the backbone for alignment?

---

> ### Author Response · Authors · 2024-07-02
> **Response by Authors**
>
> We thank the reviewer for their thoughtful comments and suggestions, which we address below.
> ## Simple Approach
> We agree with the reviewer's characterization of our approach as simple. This simplicity contrasts with prior inverse-graphics frameworks which typically rely on complex modular architectures and domain-specific inductive biases. Our approach draws inspiration from a recent shift in NLP away from task-specific designs or well-crafted supervision, toward LLMs that perform proficiently across a wide variety of tasks with relatively minor design differences and a straightforward training objective. We investigated the use of a float head for estimating continuous parameters, which enabled the application of metric supervision; however, we kept the framework deliberately generic to maintain the focus of our investigation on generalization without relying on task-specific designs.
> ## CLEVR Evaluation
> Regarding the CLEVR-CoGenT evaluation, the reviewer points out that our baseline achieves >99% accuracy on the ID condition. However, we note that in the OOD case, the model fails to generalize, with its shape-recognition accuracy dropping by 66%.
>
> We employ the ID setting primarily to validate our hypothesis that LLMs can be taught to recover precise graphics programs from demonstrations, evaluating the ability of our framework to perform comparably with domain-specific modular designs. The OOD condition, in contrast, tests a much-more challenging aspect of model capability, namely that of compositional generalization. While CLEVR is visually primitive, it serves as an established benchmark for evaluating compositional generalization. Following the CLEVR setting, we employ further evaluations to investigate generalization across other shifts, such as across parameter space and visual domains.
> ## Related Work
> We thank the reviewer for their suggestion to further improve the related-work section. However, we are unclear regarding the reviewer's comment about lacking citations. If the reviewer is aware of relevant works we may have overlooked, we would appreciate the reviewer pointing us to them so that we may incorporate them into the text.
> ## Model Terminology
> We appreciate the reviewer's comment regarding our use of the term LLM. At the time of writing, the terms LMM and LVLM lacked consistent definitions and were comparatively much-less used. By referring to our framework as IG-LLM, we intended to better highlight our exploration of inverse graphics as a language task and differentiate our work from those solving coarse semantic-level tasks. Our use of the term LLM is also in line with prior work such as Hong et al., 2023 (3D-LLM). We will add a sentence to the paper to better clarify this.

---

### Decision · Action_Editor_rCCg · 2024-07-24

**Recommendation:** Accept as is

**Comment:**

All three reviewers are in favor of acceptance. The paper presents and interesting and well-supported approach for doing inverse rendering and does extensive analysis that will be of interested to people in the community.

**Audience:**

Yes. The topics covered in the paper -- graphics, large language models, and vision as inverse rendering -- are of clear interest to many people across the computer vision community.

**Claims And Evidence:**

Yes. Post-response, all three reviewers believe the paper is supported by accurate, convincing, and clear evidence.